# Correlation Study between Quality and Sensory Characteristics of Kelp Paste by *Aspergillus oryzae* and *Aspergillus niger* during Fermentation

**DOI:** 10.3390/foods12091815

**Published:** 2023-04-27

**Authors:** Zhihong Zhong, Zhiyun Wang, Yi Zhang, Baodong Zheng, Hongliang Zeng

**Affiliations:** 1Engineering Research Center of Fujian-Taiwan Special Marine Food Processing and Nutrition, Ministry of Education, Fuzhou 350002, China; 2College of Food Science, Fujian Agriculture and Forestry University, Fuzhou 350002, China; 3Key Laboratory of Marine Biotechnology of Fujian Province, Institute of Oceanology, Fujian Agriculture and Forestry University, Fuzhou 350002, China

**Keywords:** kelp paste, *Aspergillus oryzae*, *Aspergillus niger*, fermentation, quality characteristics, sensory characteristics

## Abstract

In order to clarify the relationship between quality and sensory characteristics of kelp paste during fermentation, this study analyzed the quality and sensory characteristics of kelp paste through physicochemical indexes, nutritional components, electronic nose and electronic tongue. The results showed that with the extension of fermentation time, the contents of amino nitrogen, total acid, ammonium salt and ash increased gradually, while the pH value, moisture, fat, protein and carbohydrate decreased gradually. Short-chain alkanes such as nitrogen oxides and methane were the main causes of odor. Freshness, salinity and richness were the main indexes of kelp paste taste. Many quality indexes, such as amino nitrogen and protein, were significantly related to the odor sensor, which can better reflect the odor produced in the fermentation process of kelp paste. There was a significant correlation between quality indicators and important taste indicators such as umami, richness and salty taste, which can better reflect the taste of kelp paste during fermentation. To sum up, there was a significant correlation between the quality characteristics and sensory quality of kelp paste, so the relationship between quality characteristics and sensory characteristics in kelp paste can be clarified.

## 1. Introduction

Fermentation is a traditional food processing technology [1], and its initial purpose was mostly to store food. With the deepening of research, it was analyzed that the quality characteristics and sensory characteristics of food will change significantly after fermentation. For example, Nie et al. found that the fermentation of kelp by Monascus can effectively improve the antioxidant activity of the product and generate a variety of volatile substances that are positively correlated with good smell; Uchida et al. compared fermented laver paste with some soy paste and fish paste on the market and found that fermented laver paste was rich in total nitrogen compounds and had unique free amino acids (such as taurine), which effectively improve the unique flavor of laver paste. Therefore, fermentation technology was widely used in marine products such as fish, shrimp, shellfish and seaweed [2,3,4,5,6,7,8,9]. People improve the quality and sensory characteristics of marine fermented products through fermentation [10,11,12], but there is little research on the correlation between the quality and sensory characteristics of marine fermented products.

Kelp is a kind of renewable economic algae with rich nutrients [13,14], which has been cultivated in large quantities all over the world. By 2021, the global kelp cultivation area exceeded 2 million square kilometers, and kelp has become the largest economic algae in the world [15]. In the process of kelp production, traditional primary processing was the main way to produce instant kelp, dried kelp, kelp snacks, etc. The technology was not yet mature, and there were few deep-processed products. At present, kelp deep-processing products on the market are mostly kelp soy paste [16], but there are few paste products. Therefore, the team has developed a new type of fermented kelp paste by using its characteristics in the early stage, and they have optimized its fermentation process [17], but the research on the relationship between the quality and sensory characteristics of kelp paste during fermentation has not yet started.

Therefore, taking fermented kelp paste as the research object, this paper studied the correlation between the quality and sensory characteristics of kelp paste during the fermentation process of *Aspergillus oryzae* and *Aspergillus niger* (0 d, 10 d, 20 d, 30 d, 40 d, 50 d), including the quality indicators—physicochemical indicators and nutritional components—as well as the sensory indicators—electronic nose and electronic tongue—and further analyzed the correlation. Finally, the flavor formation law of kelp paste fermented by artificial inoculation was obtained, which laid the foundation for further development of kelp paste products.

## 2. Materials and Methods

### 2.1. Reagents and Materials

*Aspergillus oryzae* (CICC2339, spores ≥ 1.8 × 10^8^/g), and *Aspergillus niger* (CGMCC32783, spores ≥ 1.8 × 10^8^/g) were purchased from Yiyuan Kangyuan Biotechnology Co., Ltd. (Zibo, China). Salted kelp slices were purchased from Fuzhou Hailin Food Co., Ltd. (Fuzhou, China). Soybeans and salt were purchased from Fuzhou Carrefour supermarket (Fuzhou, China). A 36~38% formaldehyde solution (analytically pure) was purchased from Chengdu Kelong Chemical Reagent Factory (Chengdu, China). Sodium hydroxide (analytically pure) was purchased from Chengdu Kelon Chemical Reagent Factory (Chengdu, China).

### 2.2. Test Method

#### 2.2.1. Preparation Process of Kelp Paste

Raw material processing: the raw materials of kelp paste were treated according to the methods of Li et al. [18]. Select high-quality soybeans, and then soak them in distilled water for 12 h. The ratio of soybeans to distilled water was 1:3. After soaking, clean and drain them, and then use a wall-breaking machine to crush them. The power of the machine was 36,000 rpm for 25 s (Model HX-PB9538, Foshan Haixun Electric Appliance Co., LTD, Foshan, China). With reference to the method of Nie et al. [8], the salted kelp slices were selected and soaked in distilled water for 3 h. The ratio of kelp to distilled water was 1:3. After soaking, the sand on the kelp surface was cleaned, and after draining, it was crushed with a wall-breaking machine with a power of 36,000 rpm for 20 s (Model HX-PB9538, Foshan Haixun Electric Appliance Co., LTD, Foshan, China).

Preparation of kelp paste: put the processed kelp paste raw materials into a 500 mL beaker according to the ratio of kelp mass to soybean mass, 3:1, with the total amount of raw materials being 160 g/bottle, and then seal. Put the beaker filled with materials into the autoclave and sterilize it at 121 °C for 15 min (Model YXQ-LS-30SⅡ, Shanghai Jinghong Experimental Equipment Co., Ltd., Shanghai, China). After sterilization and cooling, add the koji in a sterile environment according to the ratio of the mass of *Aspergillus oryzae* to of *Aspergillus niger*, 2:1, then fully stir to mix the materials and seed starter evenly, then seal them and, finally, place them in a constant temperature of 34 °C for culture (Model SPX-270, Ningbo Jiangnan Instrument Factory, Ningbo, China) [8]. After 2 d of fermentation, the kelp has matured; add 75% salt water relative to the weight of the koji and stir evenly, with the salt water concentration of 14.5%. Finally, seal it and place it at 34 °C for constant temperature fermentation.

In this study, the kelp paste was divided into six groups according to the fermentation time: fermentation for 0 d (KP_0 d), fermentation for 10 d (KP_10 d), fermentation for 20 d (KP_20 d), fermentation for 30 d (KP_30 d), fermentation for 40 d (KP_40 d), fermentation for 50 d (KP_50 d), according to the fermentation time.

#### 2.2.2. Determination of Physicochemical Properties of Kelp Paste

The amino nitrogen content of the samples was measured by formaldehyde method according to Q/FHTY 0043S-2023 “Fermented Kelp Paste”. The total acid content of the samples was measured by potentiometric titration with pH meter according to Chinese standard GB 12456-2021. The pH of the samples was measured with a pH meter (Model FE28, Mettler-Tollido Instrument Shanghai Co., Ltd., Shanghai, China). The ammonium salt content of the samples was measured according to Chinese standard GB 5009.234-2016. The water content of the samples was measured by direct drying method according to Chinese standard GB 5009.3-2016.

#### 2.2.3. Determination of Basic Nutrition of Kelp Paste

The protein content of the samples was measured by Kjeldahl determination according to Chinese standard GB 5009.5-2016. The fat content of the samples was measured by Soxhlet extraction according to Chinese standard GB 5009.6-2016. The carbohydrate content of the samples was measured according to Chinese standard GB 28050-2011. The sodium content of the samples was measured by inductively coupled plasma mass spectrometry according to Chinese standard GB 5009.91-2017. The ash content of the samples was measured according to Chinese standard GB 5009.4-2016.

#### 2.2.4. Measurement of Electronic Nose

The electronic nose analysis was performed by the PEN-3 system (Airsense Analyticd Inc., Schwerin, Germany). Firstly, the injection needle was inserted into the headspace bottle containing the sample, and then the sample was determined by electronic nose system. Determination conditions: sampling time was 1 s/group. Self-cleaning time of sensor was 80 s. The sensor zeroing time was 5 s. The sample preparation time was 5 s. The injection flow was 400 mL/min. The analysis sampling time was 80 s.

#### 2.2.5. Measurement of Electronic Tongue

The TS-5000Z electronic tongue system (Super tongue, Insent Company, Tokyo, Japan) was used to evaluate the taste of kelp paste. The specific operation steps are as follows: weigh 30 g of the sample and put it into a beaker, add 150 mL of purified water, and then conduct ultrasonic treatment for 10 min, with the power of 4000 rpm centrifugation for 5 min. After ultrasonic treatment, filter it with filter paper, and then take the filtrate and put it into the special beakers for electronic tongue. Finally, put the filtrate on the electronic tongue analysis device, and test the six tastes: fresh, salty, sour, bitter, astringent and sweet.

### 2.3. Statistical Analysis

All tests were repeated three times, and the test data were expressed in the form of “mean ± standard deviation”. The variance analysis method and bivariate correlation analysis method of SPSS version 20.0 (SPSS Inc., Chicago, IL, USA) were used for difference analyses. Data processing such as discounted charts, PCA analysis and radar chart analysis were handled by Origin 2021 software (Origin Lab, Northampton, MA, USA).

## 3. Results and Analysis

### 3.1. Changes of Physicochemical Indexes of Kelp Paste at Different Fermentation Times

#### 3.1.1. Amino Nitrogen

Figure 1a showed the trend of amino nitrogen content at different fermentation times. It can be seen from the figure that the content of amino nitrogen gradually increased during the whole fermentation period. At the initial stage of fermentation, as *Aspergillus oryzae* and *Aspergillus niger* constantly produced protease to hydrolyze protein, so the content of amino nitrogen showed a significant difference from 0~10 days and reached 0.76 ± 0.07 g/100 g on the 10th day, meeting the requirements for amino nitrogen in kelp paste fermentation (≥0.5 g/100 g). With the continuous fermentation, the fermentation environment gradually became acidic in the middle and late stages, and the activities of neutral protease and alkaline protease were weakened [8,19], which made the amino nitrogen content tend to be flat in 30~50 days, and there was no significant difference between them, reaching 1.3 ± 0.13 g/100 g after the fermentation. It can be seen that fermentation for 30~50 days can effectively improve the umami components in kelp paste, and the quality of kelp paste can be greatly improved, which was consistent with the determination trend of amino nitrogen index in *Aspergillus oryzae*-fermented soy paste by Xu et al. [19].

#### 3.1.2. Total Acid and pH Value

Changes in total acid content and pH value will directly affect the fermentation process of kelp paste—which is closely related to the enzyme system produced by mold and the growth and metabolism of microorganisms—and ultimately affect the quality of kelp paste. As can be seen from Figure 1b,c, when the kelp paste was not fermented, there was almost no acid in it. With the progress of fermentation, in the initial stage of fermentation, microorganisms used sufficient proteins, carbohydrates and other substances in the environment to carry out a large number of reproduction and acid production activities, forming acidic substances such as organic acids and fatty acids [19] so that the acidity increased rapidly in 0~20 days, during which the pH value dropped rapidly; therefore the total acid and pH value at this stage were significantly different. In the middle and late stages of fermentation, due to the accumulation of acidic substances in the environment, the fermentation environment was in an acidic state to a certain extent [8,20], which weakens enzymatic hydrolysis and microbial action, and the total acid content and pH value slowly decreased. The total acid content after fermentation was 0.55 ± 0.04 g/100 g, and the pH value was 5.51 ± 0.13 g/100 g. There was no significant difference in the pH value after 30~50 days of fermentation, which indicated that the acidic environment of fermentation had basically stabilized. The pH value in this study was consistent with the determination trend of pH value in fermented kimchi by Lee et al., and the total acid was consistent with the determination trend of total acid index in soy paste by Lin et al. [21,22].

#### 3.1.3. Ammonium Salt

The ammonium salts in kelp paste are mainly produced by excessive enzymatic hydrolysis of proteins by proteases, and an appropriate amount of ammonium salts has a positive effect on the quality of kelp paste. It can be seen from Figure 1d that with the extension of fermentation time, the content of ammonium salt was generally on the rise; because a large number of proteases were produced in 0~10 days of fermentation, ammonium salt increased significantly at this stage, and the ammonium salt content at the end of fermentation was 0.17 ± 0.01 g/100 g. It can be seen from the figure that during the entire fermentation process, the ammonium salt content did not exceed 30% of the amino nitrogen content at the corresponding time point, which met the provisions of Q/FHTY 0043S-2023 “Fermented Kelp Paste”.

#### 3.1.4. Water Content

Figure 1e showed the dynamic change of moisture content during the fermentation of the kelp paste. It can be seen from the experimental result that the moisture content of the kelp paste decreased significantly during fermentation, from 76.20 ± 0.50% to 64.43 ± 0.40%. The results were consistent with the determination trend of water content in soybean paste by Tian et al., and the water content decreased significantly with the extension of fermentation time [23]. A small decrease in water content may be due to the growth and metabolic activities of microorganisms that consume some of the water, while evaporation also produces certain losses.

#### 3.1.5. Correlation Analysis of Physicochemical Indices and Fermentation Time

From the correlation analysis of physicochemical indexes and fermentation time in the fermentation process of kelp paste in Table 1, it can be seen that there was a very significant positive correlation between fermentation time and amino nitrogen, which showed that fermentation time plays an extremely important role in the formation of umami substances in kelp paste, and it was necessary to control the accumulation of ammonium salts with the extension of fermentation time.

### 3.2. Nutritional Composition Change of Kelp Paste at Different Fermentation Times

#### 3.2.1. Protein Content

As can be seen from Figure 2a, the rich substances such as fat and carbohydrate in kelp paste samples provide sufficient nitrogen and carbon sources for the growth and metabolism of *Aspergillus oryzae* and *Aspergillus niger*, respectively, so that their protease activities reach the maximum at the initial stage of fermentation. Therefore, protein can be decomposed into small molecules such as amino acids, peptides and organic acids to the maximum extent [24,25], and the content of protein decreased significantly during the 0~20 days of fermentation. In the late stage of fermentation, due to the change of fermentation environment and the decrease of energy-supplying substances, the protease activity was weakened [20,25], so the protein content gradually stabilized between 30~50 days, and there was no significant difference. After the fermentation, the protein content was 2.97 ± 0.15 g/100 g. The process improves the bioavailability of protein in raw materials and the quality of protein.

Enzymatic hydrolysis of protein will produce a variety of flavor substances that will affect the quality of kelp, such as umami peptides, amino acids, aldehydes and organic acids. Delicious amino acids such as glutamic acid, aspartic acid and glycine, and sweet amino acids such as threonine, serine and lysine all give kelp paste a unique flavor [26].

#### 3.2.2. Fat Content

As shown in Figure 2b, with the extension of fermentation time, the fat content in the kelp paste gradually decreased, indicating that fermentation would have a certain impact on the fat content. The fat content decreased significantly from 0~10 days after fermentation and then remained stable, at 1.57 ± 0.15 g/100 g, after fermentation. This was mainly due to the high efficiency of lipase in the early stage of fermentation, which can make full use of the substrate to hydrolyze the fat in raw materials to produce fatty acids, aldehydes or ketones. At the same time, under the action of halophilic bacteria, fat was decomposed to produce a variety of fatty acids beneficial to the human body, and then the oxidation reaction affects the smell and flavor of kelp paste [26,27].

#### 3.2.3. Carbohydrate Content

As shown in Figure 2c, the carbohydrate content in the kelp paste decreased with the extension of the fermentation time, which indicated that fermentation would have a certain impact on carbohydrate content. At the initial stage of fermentation, *Aspergillus oryzae* and *Aspergillus niger* secreted a large amount of glycosidase and glycosyltransferase to saccharify and hydrolyze starchy raw materials in the kelp paste, producing reducing sugar, which provided a certain material basis for the growth and metabolism of microorganisms. Therefore, the content of carbohydrates decreased significantly after 0~10 days of fermentation. In the middle stage of fermentation, the activities of glycosidase and other enzymes were weakened, while protein and fat in raw materials were still undergoing enzymatic hydrolysis and degraded into various small molecular substances, thus significantly increasing the carbohydrate content after 20 days of fermentation. In the fermentation process, carbohydrates were metabolized to produce lactic acid, acetic acid and other substances, which gives kelp paste a certain sour taste [25].

#### 3.2.4. Sodium and Ash Content

As can be seen from Figure 2d, the sodium content in the kelp paste increased significantly during the whole fermentation process and remained at about 2.63~2.64 g/100 g in general, which echoed the changing trend of water content. The main reason for the significant increase of sodium content was that with the extension of the fermentation time, the water in the kelp paste samples was lost due to microbial action and evaporation, so the sodium content increased significantly during the whole fermentation process. As can be seen from Figure 2e, the ash content increased significantly during the 0~10 days of fermentation, which may be mainly due to the rapid growth and metabolism of microorganisms in the early stage of fermentation; the volatile substances in the kelp paste were utilized, which led to the loss of dry matter, while the absolute ash content remained unchanged, so the ash content increased, which was consistent with Dai et al.’s trend in fecal fermentation [28].

#### 3.2.5. Correlation Analysis of Nutrients and Fermentation Time

From the correlation analysis of nutrients and fermentation time in the fermentation process of kelp paste in Table 2, it can be seen that with the extension of fermentation time, the contents of sodium and ash increase, while the contents of fat, protein and carbohydrate decrease; the fermentation time was only positively correlated with sodium. The results showed that *Aspergillus oryzae* and *Aspergillus niger* could effectively degrade macromolecular substances such as protein and fat during the whole fermentation process, and then form flavor components of the kelp paste such as umami peptides, amino acids and organic acids, which was beneficial for improving the quality characteristics of kelp paste.

### 3.3. Electronic Nose Analysis of Kelp Paste at Different Fermentation Times

#### 3.3.1. PCA Principal Component Analysis

The odor information in the fermentation process of kelp paste was shown in Figure 3a. The contribution rates of the first principal component (PC1) and the second principal component (PC2) were 74.59% and 16.60%, respectively, and the total contribution rate was 91.19%, indicating that the extracted odor can reflect the information of different fermentation times of the kelp paste. During the whole fermentation process, the unfermented samples were concentrated in the second quadrant and the fermented samples were mostly concentrated in the first, third and fourth quadrants, with certain overlap, indicating that the kelp paste had certain similarities in odor during different fermentation times. At the same time, according to the analysis results in Table 3, the kelp paste had certain differences in odor during different fermentation times.

#### 3.3.2. LDA Analysis

Figure 3b showed the LDA analysis diagram of the kelp paste under different fermentation times. It can be seen from the figure that the contribution rates of the first principal component (PC1) and the second principal component (PC2) were 98.27% and 1.35%, respectively, and the total contribution rate was 99.62%, which can better show the trend changes between different fermentation samples. It was observed that the unfermented samples were obviously different from the fermented samples, and the odor speed of the fermented samples showed an increasing change, with the largest change from the 0th to the 10th day. The speed change was small during the fermentation process from 10~50 days, and the odor of 40~50 days appeared to partially overlap. This change was caused by the action of enzymes and the growth and metabolism of microorganisms in the fermentation process. In the early stage of fermentation, *Aspergillus oryzae* and *Aspergillus niger* secreted various enzymes for enzymolysis. Meanwhile, the kelp paste had completed the koji-making process, and a large number of microorganisms were growing and metabolizing, so the odor inside the kelp paste changed greatly. In the middle stage of fermentation, the function of the enzymes and the energy-supplying substances of microorganisms were weakened, so that the functions of fat hydrolysis and starch saccharification were weakened. Therefore, the odor changes in the kelp paste gradually became smaller. When the fermentation time continued to extend, the number of microorganisms gradually decreased and basically stopped growing. Therefore, the odor of the samples from 40~50 days overlapped, and this time was also the key time for flavor formation.

#### 3.3.3. Loadings Analysis

Figure 3c showed the loading analysis chart of the kelp paste. The contribution rates of the first principal component (PC1) and the second principal component (PC2) were 86.29% and 10.35%, respectively, and the total contribution rate was 96.64%. It could be seen from the figure that sensor 2 has the largest contribution to the first principal component, sensor 6 had the largest contribution to the second principal component, and sensors 7, 8, and 9 had certain contributions to the first and second principal components. The response values of sensors 1, 3, 4, 5, and 10 were close to zero, indicating that the recognition function of these five sensors was weak. At the same time, according to Table 4, sensors 2 and 6 respectively correspond to nitrogen oxides and methane and other short chain alkanes, which play a major role in the odor of kelp paste during fermentation.

#### 3.3.4. Correlation Analysis between Physicochemical and Flavor Characteristics in the Fermentation Process of Kelp Paste

In order to reveal the correlation between various indexes and flavor changes in the fermentation process of kelp paste, Pearson correlation coefficient and significance were used to analyze it. As shown in Table 5, amino nitrogen and ammonium salt in the kelp paste during fermentation were negatively correlated with W1C, W5S, W3C and W2W and positively correlated with W2S and W3S. At the same time, the amino nitrogen was negatively correlated with W5C. Total acid was negatively correlated with W5S. Water content was negatively correlated with W2S and positively correlated with W3C. PH was positively correlated with W1C, W3C, W5C and W2W and negatively correlated with W2S and W3S. Protein and carbohydrate were positively correlated with W1C, W5S, W3C, W2W and negatively correlated with W3S. Fat was positively correlated with W1C, W5S and W2W. Sodium content was negatively correlated with W1C, W3C, W5C, W2W and positively correlated with W2S and W3S. Ash content was negatively correlated with W1C, W5S, W3C and W2W and significantly positively correlated with W3S. In summary, it can be seen that most quality indicators can be significantly correlated with the response values (*p* < 0.01) in W1C, W5S, W3C, W2S, W2W and W3S sensors, indicating that the quality indicators, such as amino nitrogen and total acid, can better reflect the flavor sensory quality brought by benzene, nitrogen oxides, ammonia, alcohols, ethers, aldehydes, ketones, organic sulfides and alkanes in the fermentation process of the kelp paste. There were certain similarities with the taste analysis results of Lu et al. on shrimp paste and the taste analysis results of Li et al. on seaweed [29,30].

### 3.4. Electronic Tongue Analysis of Kelp Paste at Different Fermentation Times

#### 3.4.1. PCA Principal Component Analysis

It can be seen from Figure 4a that the contribution rates of the first principal component (PC1) and the second principal component (PC2) were 60.06% and 23.96%, respectively, and the total contribution rate was 84.02%, which can better reflect the original information of the sample. As can be seen from the figure, the unfermented samples were concentrated in the first quadrant, and the fermented samples were mostly concentrated in the second, third and fourth quadrants, which indicates that the fermentation of the kelp paste can be well distinguished according to the taste information.

Table 6 showed the principal component load matrix, which mainly reflects the contribution of different taste variables to the principal component and directly reflects their contribution to the principal component according to the absolute value of the variables. The higher the relationship between variables, the closer they were. According to Table 6, it can be seen that the coefficient of the first principal component in variable saltiness, richness (fresh aftertaste) and sweetness was larger, and the coefficient of the second principal component in variable sweetness, sourness and richness was larger, indicating that these flavors were the main characteristic flavors of the fermented kelp paste.

The umami taste of kelp paste mainly comes from the fermentation of raw materials and microbial actions, such as umami amino acids, amino nitrogen, valeraldehyde, phenethyl alcohol, 2-octenal, umami peptides, etc. Therefore, umami taste was an important indicator for sensory evaluation of the kelp paste and also the main parameter for evaluating the quality [31,32,33,34]. Salty taste in the kelp paste was mainly composed of two parts; one part was rich mineral elements carried by kelp raw materials, such as Na^+^, K^+^, etc. The other part mainly came from salt water added in the fermentation process. The sour taste was mainly caused by microbial action in the fermentation process, which produced a variety of free fatty acids, organic acids and some volatile sour substances, such as butyric acid, palmitic acid, p-ethylphenol, etc. These substances added a sour taste to the kelp paste, and the sour taste was also an important index that affected the quality of the kelp paste.

#### 3.4.2. Correlation Analysis of Response Value of Electronic Tongue Sensor during Kelp Paste Fermentation

The Pearson correlation coefficient between the response values of the electronic tongue sensor was shown in Table 7. Except for the sour and bitter aftertaste, the bitter and salty tastes show a significant negative correlation. The astringency was extremely negatively correlated with the freshness, saltiness and richness, extremely positively correlated with the astringency aftertaste. It was also positively correlated with the sweetness. Bitter aftertaste was negatively correlated with astringency aftertaste, fresh taste, salty taste and sweet taste, positively correlated with richness. Fresh taste was extremely positively correlated with richness and salty taste. Richness was extremely positively correlated with salty taste and negatively correlated with sweet taste. It can be seen that the extracted response value information has a high correlation.

#### 3.4.3. Analysis Chart of Taste Radar

The taste of the kelp paste samples with different fermentation times was analyzed by electronic tongue technology, and the radar chart was drawn according to the response values of the kelp paste samples in nine sensors, as shown in Figure 4b. It can be seen from the figure that the profile changes of sour taste, bitter taste, astringent taste, bitter aftertaste and astringent aftertaste of the kelp paste samples with different fermentation times were basically the same. With the extension of fermentation time, it was found that the sour taste of all the kelp paste was below the tasteless point, and there was no obvious sour taste in the kelp paste from 0~50 days. The bitter aftertaste, astringent aftertaste and sweetness in the kelp paste were close to or lower than the tasteless point, and the umami, salty and rich tastes were obviously enhanced. Therefore, it can be concluded that umami, salty and rich tastes are effective and important taste indexes of kelp paste, and the results were related to many factors in the physicochemical indexes and nutritional components indexes.

#### 3.4.4. Correlation Analysis between Electronic Tongue Response Value and Fermentation Time

As can be seen from Table 8, the taste of the kelp paste was generally fresh and salty. As shown in Table 8, with the extension of the fermentation time, sourness, bitterness, astringency, bitter aftertaste, astringent aftertaste and sweetness weakened, umami, richness and saltiness increased, and fermentation time was significantly positively correlated with umami and richness, and extremely positively correlated with salty taste, which was consistent with the analysis results of amino nitrogen and sodium content in the quality indexes.

#### 3.4.5. Correlation Analysis between Physicochemical Indexes and Taste Characteristics in the Fermentation Process of Kelp Paste

The correlation analysis between the physicochemical indexes and sensory characteristics was shown in Table 9, which showed that there was a certain correlation between the physicochemical indexes and sensory characteristics of the kelp paste fermentation time. Bitter taste, astringent aftertaste and sweet taste had extremely significant or significant negative correlation with amino nitrogen, while umami taste, richness and salty taste had an extremely significant positive correlation with amino nitrogen, which indicates that amino nitrogen, free amino acids, peptides and other substances produced through a series of enzymatic hydrolysis and microbial growth and metabolism were important sources of the umami taste and richness of the kelp paste.

Freshness, richness and saltiness were extremely negatively correlated with pH value, while astringent aftertaste and sweetness were positively correlated with pH value. There was a significant negative correlation between sweetness and total acid. The astringent aftertaste was extremely negatively correlated with ammonium salt, the astringent taste and sweet taste were negatively correlated with ammonium salt, and the richness was positively correlated with ammonium salt. Salty taste was extremely negatively correlated with water content, fresh taste and richness were negatively correlated with water content and astringent taste was positively correlated with water content. The astringent taste was extremely negatively correlated with sodium content, the astringent aftertaste and sweet taste were negatively correlated with sodium content and the fresh taste, richness and salty taste were extremely positively correlated with sodium content. In summary, it can be seen that the amino nitrogen, pH, ammonium salt, moisture content, protein, carbohydrate, sodium and ash of the kelp paste samples have significant correlations (*p* < 0.01) on astringency, astringent aftertaste, umami, richness, saltiness and sweetness at different fermentation stages, indicating that the quality index can better reflect the sensory quality of taste in the kelp paste.

## 4. Conclusions

In this paper, the relationship between quality characteristics and organoleptic properties during the fermentation process of kelp paste was studied. The results showed that the contents of amino nitrogen, total acid, ammonium salt and ash in the quality characteristics gradually increased with the extension of fermentation time; the pH value, moisture, fat, protein and carbohydrate gradually decreased with the extension of fermentation time; and the sodium content remained basically unchanged. Electronic nose results showed that nitrogen oxides and short-chain alkanes such as methane were the main odors of kelp paste. The correlation results between quality indicators and electronic nose showed that all quality indicators except moisture content could be significantly correlated with the response values in W1C, W5S, W3C, W2S, W2W and W3S sensors, so it could better reflect the flavor sensory quality brought by benzene, nitrogen oxides, ammonia, alcohols, ethers, aldehydes, ketones, organic sulfides and alkanes in the fermentation process of kelp paste. The results of electronic tongue showed that umami, saltiness and richness were the main taste substances of kelp paste. The correlation results between quality index and electronic tongue showed that amino nitrogen, pH, ammonium salt, moisture content, protein, carbohydrate, sodium and ash of the kelp paste samples had significant correlations to astringency, astringent aftertaste, umami, richness, saltiness and sweetness, so amino nitrogen, pH, ammonium salt, moisture content, protein, carbohydrate, sodium and ash could better reflect the sensory quality of taste in kelp paste. To sum up, there was a significant correlation between the quality characteristics and sensory quality of kelp paste, so the relationship between quality characteristics and sensory characteristics in kelp paste can be clarified.

## Figures and Tables

**Figure 1 foods-12-01815-f001:**
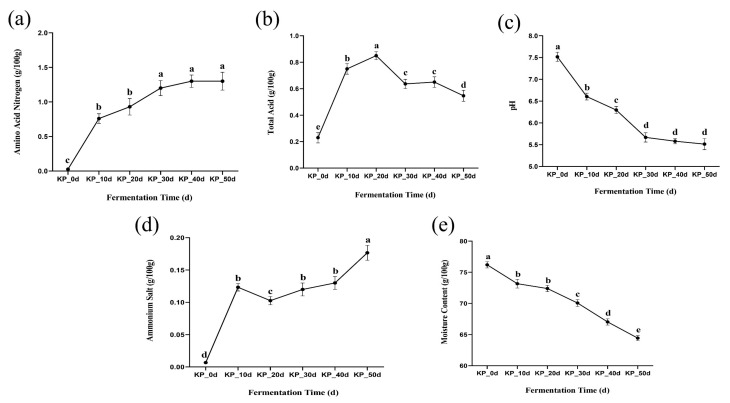
Changes of physicochemical indexes of kelp paste at different fermentation times: (**a**) changes of amino nitrogen, (**b**) changes of total acid, (**c**) changes of pH, (**d**) change of ammonium salt, (**e**) change of moisture content. Note: Statistically significant effects were assumed for *p* < 0.05. The following figure was the same. Note: Different lowercase letters represent significant differences between groups (*p* < 0.05).

**Figure 2 foods-12-01815-f002:**
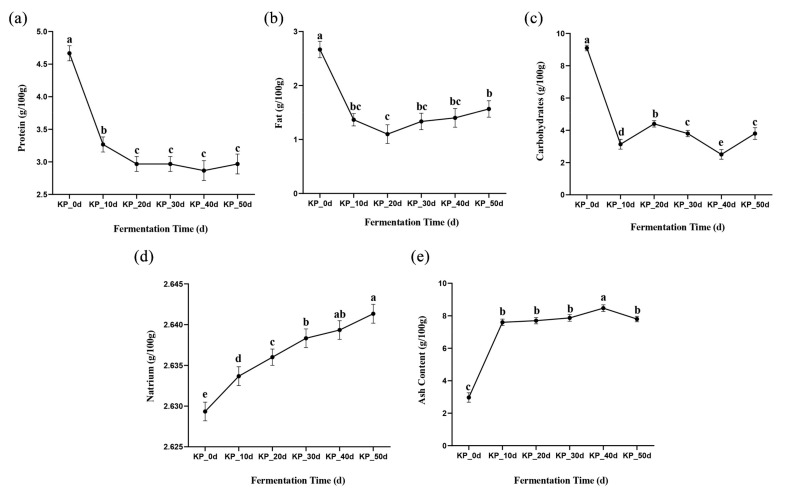
Changes in nutritional components of kelp paste at different fermentation times: (**a**) changes in protein, (**b**) changes in fat, (**c**) changes in carbohydrate, (**d**) changes in sodium, (**e**) changes in ash content. Note: Different lowercase letters represent significant differences between groups (*p* < 0.05).

**Figure 3 foods-12-01815-f003:**
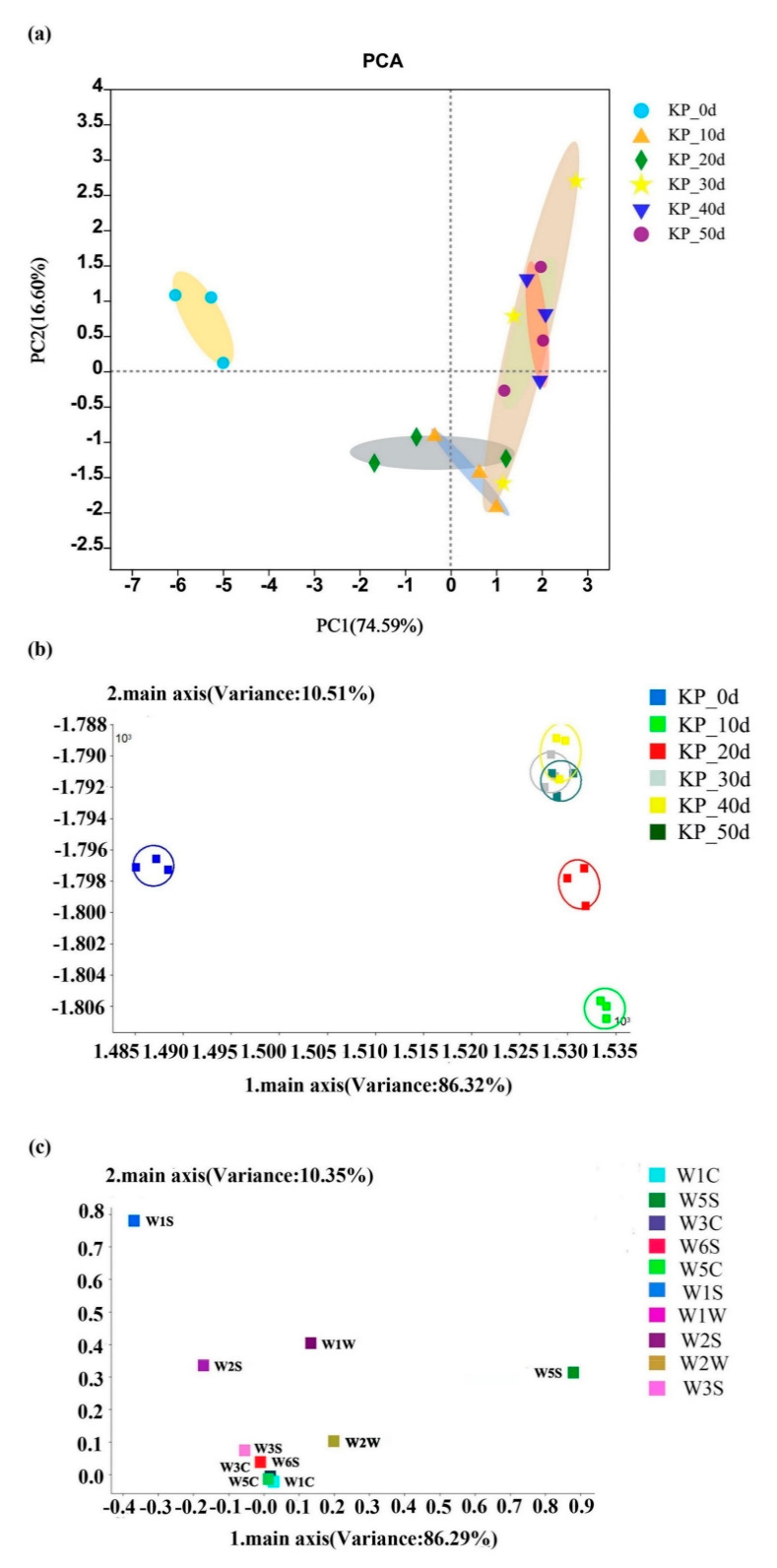
Electronic nose analysis of kelp paste at different fermentation times: (**a**) PCA analysis, (**b**) LDA analysis, (**c**) Loading analysis.

**Figure 4 foods-12-01815-f004:**
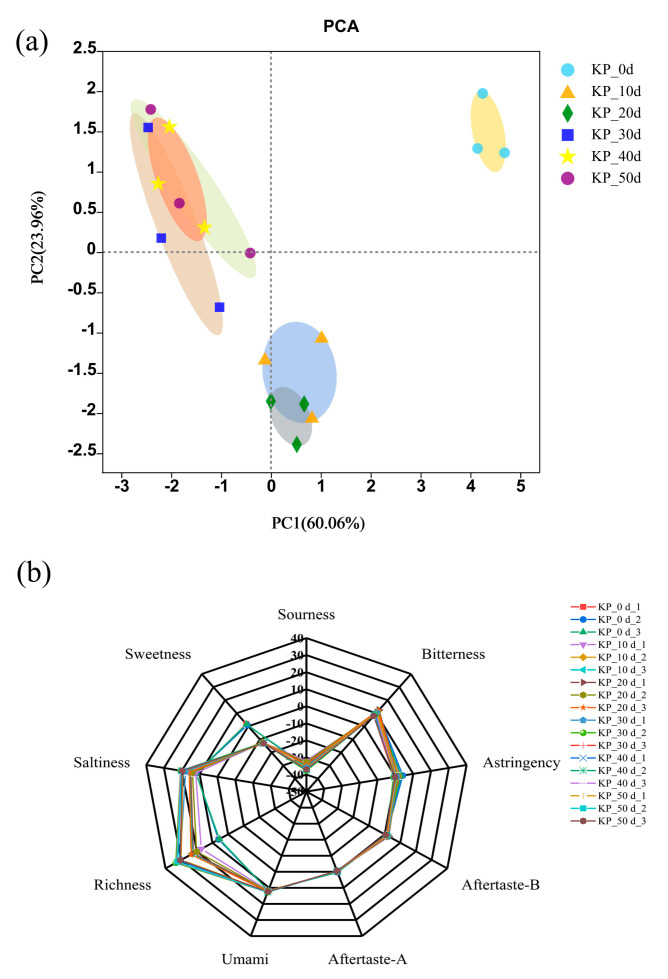
Electronic tongue analysis of kelp paste at different fermentation time: (**a**) PCA analysis, (**b**) radar analysis.

**Table 1 foods-12-01815-t001:** Correlation analysis of physicochemical index and fermentation time.

Test Items	Correlation Coefficient	Significance	Confidence
amino nitrogen	0.903	**	0.01
total acid	−0.047		0.05
pH	−0.931	**	0.01
ammonium salt	0.806	**	0.01
water content	−0.981	**	0.01

Note: ** indicates significant correlation at 0.01 level.

**Table 2 foods-12-01815-t002:** Correlation analysis of nutrition and fermentation time.

Test Items	Correlation Coefficient	Significance	Confidence
fat	−0.496		0.05
protein	−0.750		0.05
carbohydrate	−0.655		0.05
sodium	0.976	**	0.01
ash	0.709		0.05

Note: ** Indicates significant correlation at 0.01 level.

**Table 3 foods-12-01815-t003:** Table of significance analysis of electronic nose in different fermentation stages of kelp paste.

Time	KP_0 d	KP_10 d	KP_20 d	KP_30 d	KP_40 d	KP_50 d
0 d		0.979 **	0.976 **	0.984 **	0.982 **	0.979 **
10 d	0.979 **		0.487	0.548 *	0.874 *	0.968 **
20 d	0.976 **	0.487		0.501 *	0.601 *	0.550 *
30 d	0.984 **	0.548 *	0.501 *		0.079	0.289
40 d	0.982 **	0.874 *	0.601 *	0.079		0.222
50 d	0.979 **	0.968 **	0.550 *	0.289	0.222	

Note: * indicates significant correlation at 0.05 level. ** Indicates significant correlation at 0.01 level.

**Table 4 foods-12-01815-t004:** Name of electronic sensor and its response substance.

Array Serial Number	Sensor Name	Performance Description
1	W1C	aromatic components, benzene
2	W5S	high sensitivity, very sensitive to nitrogen oxides
3	W3C	ammonia, sensitive to aromatic components
4	W6S	mainly selective for hydrogen
5	W5C	alkane aromatic composition
6	W1S	sensitive to short chain alkanes such as methane
7	W1W	sensitive to inorganic sulfide
8	W2S	sensitive to alcohol ether aldehyde ketones
9	W2W	aromatic composition, sensitive to organic sulfide
10	W3S	sensitive to alkanes, long chain alkanes

**Table 5 foods-12-01815-t005:** Correlation analysis between physicochemical indexes and flavor characteristics of kelp paste.

Test Items	Amino Nitrogen g/100 g	Total Acid g/100 g	pH	Ammonium Salt g/100 g	Water Content g/100 g	Protein g/100 g	Carbohydrate g/100 g	Fat g/100 g	Sodium g/100 g	Ash g/100 g
W1C	−0.961 **	−0.691	0.930 **	−0.957 **	0.806	0.956 **	0.956 **	0.848 *	−0.900 *	−0.966 **
W5S	−0.836 *	−0.920 **	0.753	−0.859 *	0.562	0.956 **	0.950 **	0.975 **	−0.718	−0.971 **
W3C	−0.943 **	−0.509	0.943 **	−0.879 *	0.813 *	0.870 *	0.876 *	0.714	−0.892 *	−0.879 *
W6S	0.554	−0.094	−0.609	0.368	−0.698	−0.349	−0.398	−0.083	0.594	0.360
W5C	−0.883 *	−0.374	0.906 *	−0.747	0.761	0.772	0.784	0.596	−0.833 *	−0.781
W1S	0.773	0.385	−0.791	0.699	−0.593	−0.702	−0.699	−0.591	0.711	0.707
W1W	−0.149	−0.456	0.054	−0.481	0.185	0.290	0.379	0.355	−0.139	−0.333
W2S	0.926 **	0.331	−0.958 **	0.852 *	−0.891 *	−0.790	−0.753	−0.575	0.933 **	0.776
W2W	−0.902 *	−0.766	0.843 *	−0.965 **	0.762	0.943 **	0.960 **	0.869 *	−0.838 *	−0.962 **
W3S	0.923 **	0.591	−0.898 *	0.900 *	−0.792	−0.887 *	−0.953 **	−0.753	0.850 *	0.921 **

Note: * indicates significant correlation at 0.05 level. ** Indicates significant correlation at 0.01 level.

**Table 6 foods-12-01815-t006:** Electronic tongue principal component load matrix.

Taste	PC1	PC2
Sourness	−0.043319	−0.536958
Bitterness	−0.050797	−0.210642
Astringency	−0.09674	−0.109023
Aftertaste-B	0.003625	−0.131871
Aftertaste-A	−0.020697	0.012898
Umami	0.022709	0.039418
Richness	0.679391	0.482649
Saltiness	0.624122	−0.255836
Sweetness	−0.366234	0.581298
Eigenvalue	196.351363	9.557366
Contribution Rate	94.613112	4.605275

**Table 7 foods-12-01815-t007:** Correlation analysis of response value of sensor.

Items	Sourness	Bitterness	Astringency	Aftertaste-B	Aftertaste-A	Umami	Richness	Saltiness	Sweetness
Sourness	1.000 **	0.794	0.508	0.752	0.150	−0.590	−0.468	−0.716	0.023
Bitterness		1.000 **	0.783	0.451	0.533	−0.787	−0.732	−0.843 *	0.443
Astringency			1.000 **	0.005	0.918 **	−0.987 **	−0.996 **	−0.958 **	0.869 *
Aftertaste-B				1.000 **	−0.254	−0.071	0.024	−0.265	−0.383
Aftertaste-A					1.000 **	−0.870 *	−0.939 **	−0.795	0.990 **
Umami						1.000 **	0.984 **	0.977 **	−0.808
Richness							1.000 **	0.950 **	−0.893 *
Saltiness								1.000 **	−0.712
Sweetness									1.000 **

Note: * indicates significant correlation at 0.05 level. ** Indicates significant correlation at 0.01 level.

**Table 8 foods-12-01815-t008:** Correlation analysis between electron and fermentation stage.

Test Items	Correlation Coefficient	Significance	Confidence
Sourness	−0.654		0.05
Bitterness	−0.745		0.05
Astringency	−0.884	*	0.05
Aftertaste-B	−0.359		0.05
Aftertaste-A	−0.775		0.05
Umami	0.884	*	0.05
Richness	0.888	*	0.05
Saltiness	0.947	**	0.01
Sweetness	−0.687		0.05

Note: * indicates significant correlation at 0.05 level. ** Indicates significant correlation at 0.01 level.

**Table 9 foods-12-01815-t009:** Correlation analysis between physicochemical indexes and taste characteristics of kelp paste.

Test Items	Amino Nitrogen g/100 g	Total Acid g/100 g	pH	Ammonium Salt g/100 g	Water Content g/100 g	Protein g/100 g	Carbohydrate g/100 g	Fat g/100 g	Sodium g/100 g	Ash g/100 g
Sourness	−0.407	0.451	0.526	−0.234	0.640	0.111	−0.199	0.060	−0.531	−0.074
Bitterness	−0.662	0.077	0.728	−0.634	0.777	0.440	0.171	0.512	−0.704	−0.461
Astringency	−0.977 **	−0.507	0.983 **	−0.885 *	0.867 *	0.893 *	0.720	0.858 *	−0.946 **	−0.885 *
Aftertaste-B	0.038	0.733	0.066	0.054	0.400	−0.315	−0.600	−0.297	−0.149	0.331
Aftertaste-A	−0.959 **	−0.781	0.911 *	−0.937 **	0.767	0.989 **	0.911 *	0.968 **	−0.883 *	−0.993 **
Umami	0.959 **	0.444	−0.977 **	0.809	−0.855 *	−0.854 *	−0.666	−0.809	0.931 **	0.840 *
Richness	0.991 **	0.559	−0.990 **	0.891 *	−0.867 *	−0.923 **	−0.762	−0.877 *	0.955 **	0.911 *
Saltiness	0.928 **	0.284	−0.968 **	0.794	−0.925 **	−0.770	−0.530	−0.709	−0.951 **	0.745
Sweetness	−0.915 *	−0.849 *	0.854 *	−0.912 *	0.677	0.985 **	0.951 **	0.966 **	−0.819 *	−0.992 **

Note: * indicates significant correlation at 0.05 level. ** Indicates significant correlation at 0.01 level.

## Data Availability

Data is contained within the article.

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
