# Peer review of "Correlation Study between Quality and Sensory Characteristics of Kelp Paste by Aspergillus oryzae and Aspergillus niger during Fermentation"

_foods, 2023, doi:10.3390/foods12091815_

Round 1

Reviewer 1 Report

The manuscript is of good quality and the format of figures are standard. The methods are appropriate and properly conducted.I read related references, and from my own view, this manuscript needs to revise the Methods and resluts/discussion part. 

1. Do you use which kinds, model number and bland of analysis equipments for "2.2.4. Measurement of Electronic Nose" and "2.2.5. Measurement of Electronic Tongue". 

2. All figures can be modified for readers to understand clearly. 

3. You can show or discuss the digestibilities of protein, carbohydrate, and lipids. 

Author Response

Dear Editor,

A revision on foods-2218848 has been carried out. Replies to the reviewers’ comments are listed below and the corresponding corrections were made in the revised manuscript.

Q1: Do you use which kinds, model number and bland of analysis equipments for "2.2.4. Measurement of Electronic Nose" and "2.2.5. Measurement of Electronic Tongue".

A1: Comment has been taken into account. It was supplemented accordingly. (Lines 118,125)

Q2: All figures can be modified for readers to understand clearly.

A2: Comment has been taken into account. All figures in the article were modified according to the comment .

Q3: You can show or discuss the digestibilities of protein, carbohydrate, and lipids.

A3: Comment has been taken into account. It was supplemented accordingly.  (Lines 207-209, 214-218, 223-228, 232-235, 240-242)

Finally, we appreciate a lot for the kind reviews and lots of positive suggestions from the editor and reviewers. The manuscript has been resubmitted to your journal. We look forward to your positive response. If you have any question regarding the manuscript, please contact Hongliang Zeng.

Address: Fujian Agriculture and Forestry University, College of Food Science, Fuzhou, Fujian, P. R. China 350002.

E-mail address: zhlfst@fafu.edu.cn.

Sincerely,

Hongliang Zeng

Reviewer 2 Report

Throughout the article, the information provided is not clear making it difficult for the reader to understand the purpose of this study. When reading this article, one gets the feeling that the authors limited themselves to using different statistical techniques, but without really understanding what they were doing.  With regard to the interpretation of the results  the authors limit themselves to a very simplistic reading of the results. All analyses and results must be well founded and explained so that the reader understands what is being done. 

Author Response

Dear Editor,

A revision on foods-2218848 has been carried out. Replies to the reviewers’ comments are listed below and the corresponding corrections were made in the revised manuscript.

Q1: Throughout the article, the information provided is not clear making it difficult for the reader to understand the purpose of this study. When reading this article, one gets the feeling that the authors limited themselves to using different statistical techniques, but without really understanding what they were doing. With regard to the interpretation of the results the authors limit themselves to a very simplistic reading of the results. All analyses and results must be well founded and explained so that the reader understands what is being done.

A1: Opinions have been taken into account. It has been modified accordingly.

① The missing data grouping information in the article and the legend in picture 3 have been completed.(Lines 96-99)

② Combined with the supplementary information, the data of the article were analyzed in depth, so the research purpose of this article could be clearly expounded, that was, the quality characteristics and sensory characteristics of kelp paste were studied, and the changing rules of quality characteristics and sensory characteristics of kelp paste during fermentation were clarified.Specific analysis and discussion have been marked.(Lines 11-24, 191-199, 207-209, 214-218, 223-228, 232-235, 240-242, 261-268, 346-353, 430-435, 462-466, 470-491)

 Finally, we appreciate a lot for the kind reviews and lots of positive suggestions from the editor and reviewers. The manuscript has been resubmitted to your journal. We look forward to your positive response. If you have any question regarding the manuscript, please contact Hongliang Zeng.

Address: Fujian Agriculture and Forestry University, College of Food Science, Fuzhou, Fujian, P. R. China 350002.

E-mail address: zhlfst@fafu.edu.cn.

Sincerely,

Hongliang Zeng

Reviewer 3 Report

These are my suggestions for the paper foods-2218848:

- Abstract is too long, please revise it in better manner, try to have maximum 200 words

- Expand the Introduction with details from references 1-12.

- line 52 replace Niger with niger

- lines 60-61: What are 3.042 and 3.758, and why these words are italic? Also, the concentration of spores needs to be indicated as X*10x spores/g - revise that in the paper

- Description of the preparation of the fungal suspension is required. Also, manipulation and storage techniques must be explained. 

-  The letters on graphs need to have legend and explanation above the figure name

- ALL figures must be in better resolution, especially Figure 4.

Author Response

Dear editor:

A revision on foods-2218848 has been carried out. Replies to the reviewers’ comments are listed below and the corresponding corrections were made in the revised manuscript.

Q1: Abstract is too long, please revise it in better manner, try to have maximum 200 words.   

A1: Opinions have been taken into account. The abstract has been modified accordingly. (Lines 11-24)

Q2: Expand the Introduction with details from references 1-12.

A2: Comment has been taken into account. It was revised accordingly. (Lines 29-37)

Q3: line 52 replace Niger with niger.

A3: Comment has been taken into account. It was revised accordingly.(Line 56

Q4:lines 60-61: What are 3.042 and 3.758, and why these words are italic? Also, the concentration of spores needs to be indicated as X*10x spores/g - revise that in the paper.

A4: ①3.042 and 3.758 are not internationally recognized strain numbers, so they have been deleted in this paper.

② Because the translations of Aspergillus oryzae and Aspergillus niger belong to Latin, the fonts need to be italicized.

③ The spore number has been revised according to the expert's suggestion (Lines 64-65).

Q5:Description of the preparation of the fungal suspension is required. Also, manipulation and storage techniques must be explained.

A5: The two fungi used in this study were purchased from Yiyuan Kangyuan Biotechnology Co., Ltd., which were explained in 2.1 experimental materials. According to the instructions, the storage conditions of the two fungi were sealed at room temperature for 6-8 months.(Line 65)

Q6: The letters on graphs need to have legend and explanation above the figure name.

A6: Comment has been taken into account. It was revised accordingly in Figure 3.At the same time, the sample grouping in the article was supplemented in 2.2.1 preparation technology of kelp paste(Lines 96-99).

Q7: ALL figures must be in better resolution, especially Figure 4.

A7: Comment has been taken into account. there was revised accordingly.

Finally, we appreciate a lot for the kind reviews and lots of positive suggestions from the editor and reviewers. The manuscript has been resubmitted to your journal. We look forward to your positive response. If you have any question regarding the manuscript, please contact Hongliang Zeng.

Address: Fujian Agriculture and Forestry University, College of Food Science, Fuzhou, Fujian, P. R. China 350002.

E-mail address: zhlfst@fafu.edu.cn.

Sincerely,

Hongliang Zeng

Round 2

Reviewer 2 Report

In my opinion the article as serious flow both in terms of statistical analysis and results.

The statistical analysis  section needs to be rewrite. As a I mentioned before,the information given here is sparce.  What were the statistical techniques used?  The data are suitable for these techniques, that is, the assumptions of the techniques used were verified? The only information that authors give is:

Use … to conduct statistical  analysis on the data. … and SPSS 25.0 software was  used for difference analysis.

The information provided is not clear. All analyses and results must be well founded and explained so that the reader understands what is being done.

It should be noted that the measure of central tendency is usually called mean not average. With regard to decision-making, it is usually written: “Statistically significant effects were assumed for p < 0.05.”

In the results section the authors only present the graphical representations for each of the variables without giving information about the result of the statistical analysis. Where are these results? What was the post hoc test used? Why did they not show mean values ​​and standard deviations? These measures are used to summarize the data under study.

Wasn't the goal to see if there were significant differences between the different days analyzed? If so, If so, why don't the authors talk about the conclusions of the analyzes.

Section 3.1.5

In table table 1 authors presented correlation between Physicochemical Indeces and Fermentation Time. I amk confuse isn´t fermentation time a nominal variable with six groups (d(KP_0 d), d(KP_10 d), ….50 d(KP_50 d) )? How did you determine the correlation between a nominal and a continuous variable? Please explain how?

Section 3.2.1.

What was the statistical analysis used? Where are the statistical results?

Section 3.2.5.

As I said in section 3.1.5, I don't understand how they calculate this coeficiente

Section 3.3

The information provided is not clear making it difficult for the reader to understand the reasoning behind the techniques used and the conclusions taken.

Authors should have taken care to verify the document before sending it back to reviewers. There are several sentences in which the spacing before or after is not correct.

Subsection 2.2.1

Please correct the spacing

Line 75: a space is missing at the beginning of the sentence.

Line 79: a space is missing at the beginning of the sentence.

Line 84: a space is missing after “paste:”

Line 85: please correct “3:1”

……

Line 105: correct  spacing “Ltd,Shanghai) .The ammonium”

Line 106: correct  spacing “5009.234-2016.The water”

….

Sections: 2.2.3, 2.2.4, 2.2.5,

Line 130. electronic tongue(). to correct

Line 135: When referring to descriptive measures we write “mean” not “average”

Author Response

Dear Editor,

        A revision on foods-2218848 has been carried out. Replies to the reviewers’ comments are listed below and the corresponding corrections were made in the revised manuscript.

        Q1: The statistical analysis  section needs to be rewrite. As a I mentioned before,the information given here is sparce.  What were the statistical techniques used?  The data are suitable for these techniques, that is, the assumptions of the techniques used were verified? The only information that authors give is:

        Use … to conduct statistical  analysis on the data. … and SPSS 25.0 software was  used for difference analysis.

        The information provided is not clear. All analyses and results must be well founded and explained so that the reader understands what is being done.

        A1: Comment has been taken into account. The statistical analysis of 2.3 has been modified, and the results and analysis has been supplemented(Lines 133-137, 143-155, 160-173, 179-186, 189-193, 200, 215-222, 232-233, 246-250, 254-265).

        Q2: It should be noted that the measure of central tendency is usually called mean not average. With regard to decision-making, it is usually written: “Statistically significant effects were assumed for p < 0.05.”

        A2: Comment has been taken into account. It has been modified accordingly(Lines 134, 200).

        Q3: In the results section the authors only present the graphical representations for each of the variables without giving information about the result of the statistical analysis. Where are these results? What was the post hoc test used? Why did they not show mean values and standard deviations? These measures are used to summarize the data under study. Wasn't the goal to see if there were significant differences between the different days analyzed? If so, If so, why don't the authors talk about the conclusions of the analyzes.

        A3: Comment has been taken into account. It has supplemented the analysis methods in the 2.3 statistical analysis section and the corresponding significance, mean and standard deviation analysis in the results and analysis section(Lines 143-155, 160-173, 179-186, 189-193, 215-222, 232-233, 246-250, 254-265). We appreciate a lot for your positive comment to improve the quality of the manuscript.

        Q4: In table 1 authors presented correlation between Physicochemical Indeces and Fermentation Time. I amk confuse isn´t fermentation time a nominal variable with six groups (d(KP_0 d), d(KP_10 d), ….50 d(KP_50 d) )? How did you determine the correlation between a nominal and a continuous variable? Please explain how?

        A4: Comment has been taken into account. As we known, the physicochemical factors in kelp paste was changed with fermentation time. In this paper, the construction of the correlation between physicochemical factors and fermentation time was reasonable and meaningful. And it was analyzed by SPSS bivariate correlation analysis, and this method was used to analysis the correlation of other indicators. It has been supplemented in 2.3 statistical analysis(Lines 134-135).

        Q5: Section 3.2.1., What was the statistical analysis used? Where are the statistical results?

        A5: The data analysis methods used in this part were the one-way ANOVA test of SPSS and the mean values±standard deviations. All relevant statistical results were supplemented and analysed in accordance with the recommendations(Lines 215-222, 232-233, 246-250, 254-265).

        Q6: Section 3.2.5., As I said in section 3.1.5, I don't understand how they calculate this coeficiente.

        A6: In this paper, the correlation between physicochemical factors and fermentation time was analyzed by SPSS bivariate correlation analysis, and the correlation analysis of other indicators adopted this method. The article has been supplemented in 2.3 statistical analysis(Lines 134-135).  We appreciate a lot for your positive comment to improve the quality of the manuscript.

        Q7: Section 3.3, The information provided is not clear making it difficult for the reader to understand the reasoning behind the techniques used and the conclusions taken.

        A7: Comment has been taken into account. The electronic nose data information has been supplemented in the 2.3 statistical analysis section. In this part, the electronic nose was used to detect the flavor of kelp paste. And PCA analysis, LDA analysis, Loading analysis and correlation analysis of kelp paste data were analyzed, mainly in order to discussed the flavor changes of kelp paste at different fermentation stages.  The correlation between flavor and physicochemical indeces was constructed(Lines 133-137).

        Q8: Please correct the spacing

               Line 75: a space is missing at the beginning of the sentence.

               Line 79: a space is missing at the beginning of the sentence.

               Line 84: a space is missing after “paste:”

               Line 85: please correct “3:1”

                ……

                Line 105: correct  spacing “Ltd,Shanghai) .The ammonium”

                Line 106: correct  spacing “5009.234-2016.The water”

                ….

                Sections: 2.2.3, 2.2.4, 2.2.5,

                Line 130. electronic tongue(). to correct

                Line 135: When referring to descriptive measures we write “mean” not “average”

        A8: Comment has been taken into account. The whole text has been revised accordingly. 

        Finally, we appreciate a lot for the kind reviews and lots of positive suggestions from the editor and reviewers. The manuscript has been resubmitted to your journal. We look forward to your positive response. If you have any question regarding the manuscript, please contact Hongliang Zeng.

        Address: Fujian Agriculture and Forestry University, College of Food Science, Fuzhou, Fujian, P. R. China 350002.

        E-mail address: zhlfst@fafu.edu.cn.

        Sincerely,

        Hongliang Zeng

Reviewer 3 Report

/

Author Response

Dear Editor,

        A revision on foods-2218848 has been carried out. Replies to the reviewers’ comments are listed below and the corresponding corrections were made in the revised manuscript.

        Q1: Extensive editing of English language and style required.   

        A1: Comment has been taken into account. The whole text has been revised accordingly.

        Finally, we appreciate a lot for the kind reviews and lots of positive suggestions from the editor and reviewers. The manuscript has been resubmitted to your journal. We look forward to your positive response. If you have any question regarding the manuscript, please contact Hongliang Zeng.

        Address: Fujian Agriculture and Forestry University, College of Food Science, Fuzhou, Fujian, P. R. China 350002.

        E-mail address: zhlfst@fafu.edu.cn.

        Sincerely,

        Hongliang Zeng
